# Innovative Non-Pharmacological Management of Delirium in Persons with Dementia: New Frontiers for Physiotherapy and Occupational Therapy?

**DOI:** 10.3390/geriatrics8020028

**Published:** 2023-02-22

**Authors:** Christian Pozzi, Verena C. Tatzer, Cornelia Strasser-Gugerell, Stefano Cavalli, Alessandro Morandi, Giuseppe Bellelli

**Affiliations:** 1Centre of Competence on Ageing, University of Applied Sciences and Arts of Southern Switzerland SUPSI, 6928 Manno, Switzerland; 2Public Health, University of Milano-Bicocca, 20126 Milano, Italy; 3Department of Occupational Therapy, University of Applied Sciences Wiener Neustadt, 2700 Wiener Neustadt, Austria; 4Azienda Speciale “Cremona Solidale”, 26100 Cremona, Italy; 5Parc Sanitari Pere Virgili, Vall d’Hebrón Institute of Research, 08016 Barcelona, Spain; 6School of Medicine and Surgery, University of Milano-Bicocca, 20126 Milano, Italy; 7Acute Geriatric Unit, San Gerardo Hospital, ASST-Monza, 20900 Monza, Italy

**Keywords:** delirium, dementia, rehabilitation, occupational therapy, physiotherapy

## Abstract

Background: Delirium and dementia are two of the most common geriatric syndromes, which requires innovative rehabilitation approaches. Aim: We aimed at determining which occupational therapy and physiotherapy interventions are applied with older people with delirium and dementia in different care settings. We also identified the assessment tools that were used. Materials and methods: We conducted a literature search for scientific articles published from 2012 to 2022 (PubMed, MEDLINE, AMED and CINAHL) with adults aged >65 years including experimental study designs with randomized or non-randomized intervention, exploratory studies, pilot studies, quasi-experimental studies, case series and/or clinical cases. Studies that did not use interventions that could be classified as occupational therapy or physiotherapy were excluded. Results: After applying the exclusion criteria, 9 articles were selected. The most widely used assessment to define dementia was the MMSE (N = 5; 55.5%), whereas the CAM (N = 2; 22.2%), CAM-ICU (N = 2; 22.2%) and RASS (N = 3; 33.3%) were the most widely used to define delirium. The rehabilitation interventions that were most frequently performed were early mobilization, inclusion of the caregiver during treatment, modification of the environment to encourage orientation and autonomy, the interprofessional systemic approach and engaging persons in meaningful activities. Conclusions: Despite the growing evidence on its effectiveness, the role of physiotherapy and occupational therapy interventions in the prevention and treatment of people with dementia and delirium is still emerging. More research is needed to investigate if effective occupational therapy programs known to reduce the behavioral and psychological symptoms in people with dementia are also useful for treating delirium and specifically delirium superimposed on dementia. Regarding physiotherapy, it is crucial to know about the amount and timing of intervention required. Further studies are needed including older adults with delirium superimposed on dementia to define the role of the interprofessional geriatric rehabilitation team.

## 1. Introduction

Delirium and dementia are two of the most common causes of cognitive impairment in older people. Delirium is a clinical syndrome characterized by disturbances in consciousness, cognitive function or perception, with acute onset and course that develops over a short period of time and tends to fluctuate [1]. It is a serious condition which is associated with several adverse outcomes [2]. Dementia is a neurodegenerative disease, characterized by a chronic and progressive decline in one or more cognitive domains that can interfere with the persons’ independence in daily activities [3]. Yet, although these conditions are distinct, they frequently co-exist, in this case being labeled as delirium superimposed on dementia (DSD).

To date, pharmacological treatments have shown to be ineffective for delirium prevention and treatment [4]. Moreover, systematic reviews and guidelines strongly recommend avoiding drugs active on the central nervous system, such as neuroleptics and benzodiazepines, in people at risk for developing delirium [5]. There is increasing evidence that non-pharmacological treatments are effective, especially multimodal non-pharmacological approaches provided by a multidisciplinary team [6,7,8]. The multimodal intervention adopted includes reorientation, drug reconciliation and the reduction in psychoactive drugs, the promotion of sleep, early mobilization, adequate hydration and nutrition, and the use of vision and hearing devices [2,6]. An interdisciplinary team involving geriatricians or other medical clinicians, nurses, physiotherapists, occupational therapists, speech therapists, nutritionists, clinical pharmacists and social workers should carry out this multicomponent intervention [8]. In many multimodal approaches for persons with delirium or DSD, the role, specific activities and interventions of physiotherapists and occupational therapists are emerging and thus still unclear.

The aim of this review is to explore the literature regarding the methods used to assess the cognitive, functional status and quality of life and to describe the rehabilitative interventions adopted in the studies involving occupational therapy (OT) and physical therapy (PT) for the prevention and treatment of older patients with delirium, dementia and DSD across various settings of care.

## 2. Materials and Methods

For this scoping review, we focused on the studies that examined the role of OT and PT in preventing and treating delirium and DSD in older adults. The following research questions were asked: 1. Which assessment tools were used to screen for delirium and dementia? 2. Which tools were used to assess the functional status and the quality of life of both the persons with delirium and dementia and their caregivers (outcomes)? 3. Which kind of interventions were performed during OT and PT during the rehabilitative sessions?

We searched in the scientific database PubMed and the database CINAHL, MEDLINE and AMED (using the EBSCO search mask) using a combination of keywords: physiotherapy OR physical therapist OR pt AND delirium AND dementia; delirium AND prevention OR intervention OR treatment OR program AND physiotherapy OR physical therapist OR pt; occupational therapy OR occupational therapist AND delirium AND dementia; delirium AND prevention OR intervention OR treatment OR program AND occupational therapy OR occupational therapist OR ot.

Studies were included if they were experimental (randomized or non-randomized intervention studies, qualitative study nested in a randomized control trial (RCT—mixed methods), exploratory studies, pilot studies, quasi-experimental studies, case series and/or clinical case studies) and if the mean age of subjects was ≥65 years. Articles that did not include specific OT or PT classifiable interventions and reviews were excluded. In addition, articles not written in English and published in nonpeer-reviewed journals were excluded.

We decided to limit our literature search only to the studies published in the last decade (from January 2012 to July 2022) since these publications accounted for the majority (58.45%) of all those published on this topic from the beginning up to the present. Below is Figure 1 describing the flowchart of the study.

## 3. Results

The original search yielded a total of 326 articles, which, after applying the exclusion criteria, were finally reduced to 9 studies. Table 1 summarizes their main findings.

### 3.1. Main Characteristics of the Selected Studies

The global number of patients enrolled across studies was 887, ranging from a minimum of 6 to a maximum of 370 individuals, depending on the study. Two studies were conducted in an orthogeriatric unit [9,11], two studies in ICU [12,13] and two in an acute care for the elderly (ACE) unit [10,14]. In addition, one study was conducted in a rehabilitation ward [15], one in nursing home [16] and one in a hospice [17]. Four enrolled patients with dementia, delirium [14,17] or DSD [15,16], three studies [10,12,13] enrolled patients with a mix of medical conditions, including heart failure, sepsis, infection, chronic or acute respiratory diseases, gastrointestinal problems, renal or liver failure and hemorrhage, and two studies [9,11] enrolled patients with hip fracture. The designs of the studies differed: two were RCTs [10,13], two case series [15,17], one was a qualitative study nested in an RCT [9], one a single-blind RC pilot trial [11], one a retrospective longitudinal study [12], one a prospective observational study [14] and one a sperimental feasibility study without control group [16].

### 3.2. Tools to Assess Delirium, Dementia or DSD and Assess Functional Status, Quality of Life of Patients and Caregivers

Several tools were used to assess delirium and dementia. Four studies [10,11,12,13] used either the confusion assessment method (CAM) [18], the 3D CAM [19], the short CAM or the CAM-ICU [20], three studies [12,15,16] used either the Richmond Agitation and Sedation Scale (RASS) [21] or the modified RASS [22] and one study [16] used the recognizing acute delirium as part of your routine (RADAR) [23] and Delirium-O-Meter [24] and one study [13] used the delirium rating scale [25]. Furthermore, three others did not report using any assessment tool. The Mini Mental State Examination (MMSE) [26] was used in almost all studies to evaluate the presence of dementia [9,10,13,14,15,16], alone or in combination with the Clinical Dementia Rating Scale (CDR) [27], while in other two studies [11,17] the authors used the Montreal Cognitive Assessment (MoCA) [28], alone or in combination with the Informant Questionnaire on Cognitive Decline in the Elderly (IQCODE) short form [29]. One study [12] did not report the use of any tool to assess cognition.

The Barthel Index (BI) [30] was the most commonly used tool to assess functional status [10,12,14,15,16], followed by handgrip strength [10,13] and the Tinetti Scale [31], which were used in two different studies [15,16] to assess handgrip strength and gait and balance, respectively. The Short Physical Performance Battery (SPPB) [32], the modified Iowa level of care scale [33], the Functional Independence Measure (FIM) [34] and the gait speed (m/s) were each used once in four studies [10,11,13].

Qualitative interviews with caregivers and staff were used in two studies [8,16] whereas the EuroQol Visual Analogue Scale (VAS) [9] and the EuroQol questionnaire 5 dimensions (EuroQol 5D) [34] were applied in other two studies [9,12]. In addition, two scales measured the caregiver’s perception of occupational performance with the Canadian Occupational Performance Measure (COPM) [14,35] and one study [12] used the Geriatric Depression Scale (GDS) [36] to assess symptoms of depression in persons with dementia.

### 3.3. Rehabilitation Interventions and Results

The rehabilitation interventions were heterogeneous. Most studies in PT reported either the early mobilization or motoric exercises as the core elements of the interventions [9,10,11,12,13], while one study used recumbent cycling at bedside in addition to the usual physiotherapy procedures [11]. However, the study by Martinez Velilla et al. was the only one to report specific indications for the rehabilitation processes, including the types (resistance, balance and gait training, stand-up from a chair, leg press and bilateral knee extension), the timing (two sessions a day) and the doses (2 to 3 sets of 8–10 repetitions with 30–60% maximum load equivalent) of the exercises [10].

By looking at the studies that included an OT intervention, Alvarez et al. reported the use of polysensorial stimulations (i.e., lights, tactile body stimulation) and specific activities that were perceived as meaningful by the patient [13]. A similar approach with a focus on the meaningful activities for the patient is also described in the studies by Pozzi et al. [16], Goonan et al. [14] and Bolton and Loveard [17]. Environment changes to improve a patient’s participation, sleep and orientation are described in three studies [14,15,16], while active caregiver’s involvement in the sessions was reported in most studies [10,14,15,16,17]. Staff education was provided in the studies by Goonan et al. and Bolton and Loveard [14,17].

The lack of control groups and randomization do not allow conclusions to be drawn on the generalization of efficacy of the interventions in most studies. However, physiotherapy and occupational therapy for the older people with delirium or delirium superimposed on dementia (DSD) can be assumed:

-Person-tailored exercises at a moderate intensity can improve motor performance, autonomy, cognitive functions and quality of life, but not delirium incidence [10].-Early mobilization can reduce 30-day readmission, falls, pressure sores and respiratory adverse events in ICU patients. Moreover, these interventions can improve independence in activity of daily living but cannot shorten the length of stay [12].-OT intervention is feasible [14,15,16,17] in different care setting, can improve function [13], can decrease the duration and incidence of delirium [16], can reduce behavioral disorders [14,17] and can favor patient’s discharge to home [15].

## 4. Discussion

This review aimed at exploring the literature regarding the methods used to assess cognitive and functional status and quality of life and the types of physiotherapy and occupational interventions carried out for the prevention and treatment of patients with delirium, dementia and DSD across various settings of care.

An important insight is that we only found 9 studies that addressed these issues. Although our literature search was limited to the articles published during the last 10 years in the commonly used databases and did not start from their inception, we found very few such studies. The role of both PT and OT is universally acknowledged as crucial to prevent and manage delirium, dementia and DSD, but, in fact, it is still undervalued, suggesting that obstacles in delirium care provision have a cultural basis [8]. Scientific literature suggests that the best functional outcomes are achieved through the involvement of the entire medical, nursing and rehabilitation team [35]. The power of an interdisciplinary approach lies in an accurate identification and management of all the complex issues that are associated with delirium and dementia, including the transitional care from hospital to community; a single intervention does not achieve the same results as multimodal interventions, as is commonly seen in geriatric fields [36,37]. An interdisciplinary statement of five scientific societies (European Delirium Association EDA, European Academy of Nursing Science EANS, European Geriatric Medicine Society EuGMS, Council of Occupational Therapists for European Countries COTEC, International Association of Physical Therapists working with Older People of the World Confederation for Physical Therapy IPTOP/WCPT) for the advancement of delirium care across Europe [8] clearly pointed out that health professionals should synergistically cooperate since teamwork is the cultural cornerstone of geriatric medicine.

A finding of our review is the great heterogeneity in the patients’ characteristics across studies and in the methods used to assess cognition, function and outcome measures. Only a few studies explicitly enrolled patients because they had delirium and/or dementia as the main diagnosis [14,15,16,17], while others enrolled patients with hip fracture or other conditions that might be but are not necessarily associated with delirium development and/or dementia [9,11]. The patient samples were small, and thus, we cannot draw conclusions about the efficacy of the interventions. Only one third of the selected studies had a cohort of more than 100 patients, and only two studies had a randomized controlled design [10,13]. In addition, the methods used to evaluate delirium, dementia, DSD, functional performance and the quality of life of both patients and caregivers widely differed across studies, making it hard to compare them. Furthermore, most studies did not state whether these scales were administered during OT and PT, or if these professionals were previously trained in delirium and dementia fields.

The interventions described in the selected articles were also markedly heterogeneous: early mobilization seemed to have been used in most studies in PT as well as in training of activities of daily living (ADL) as a generic approach in OT. It is of interest that both studies with an RCT design found functional improvement at the end of the trial while only one found a decrease of delirium incidence. A possible explanation is that in the study by Alvarez et al., the reduction of delirium incidence was the main outcome measure, while it was a secondary outcome measure in the study by Martinez Velilla. It should also be considered that in the study by Alvarez et al., the enrolled patients received standard strategies of delirium prevention plus occupational therapy, while in the study by Martinez Velilla et al. the patients received only physiotherapy and underwent active exercises, but no delirium prevention strategies. Furthermore, the patients enrolled in the study by Alvarez et al. were critically ill, while they were relatively stable in the study by Martinez Velilla.

Other studies found a reduction of behavioral disorders or agitation by providing a person-centered approach (occupational story, tailored and meaningful activities), modifications of the environment and the active involvement of caregivers in rehabilitation sessions [14]. One of the keys regarding occupational therapy and physical therapy interventions for the prevention and care of the person with delirium and dementia is the importance given to the therapeutic environment and the active involvement of the family caregivers in the care process. Keeping that in mind, we think that a personalization of the environment should be encouraged to improve a patient’s rest, circadian sleep rhythm and orientation since they can be useful to prevent delirium, as suggested by international guidelines and systematic reviews [5]. Accordingly, we recommend professionals who look after those patients with delirium and dementia encourage them to bring some personal items into the ward (i.e., photographs, alarm clocks, watches) [38,39,40,41]. The ward should also offer an environment to improve the patient’s orientation (i.e., calendar), active movement (handrails and a room equipped for physical rehabilitation) and occupational recovery (i.e., an occupational therapy room). Indeed, keeping older adults engaged in meaningful activities is crucial, but often overlooked [42]. Modern hospitals should additionally encourage a patient’s active exercise and movement to avoid the onset of hospital-acquired disability, well described by Covinsky et al. [43].

Some studies supported the active involvement of the caregiver in the PT and OT sessions, assuming that this could facilitate the patient’s functional and occupational performance. However, the studies did not describe in detail when and how caregivers were involved. There is evidence that caregivers can be successfully integrated in the programs for the management of people with dementia at home in occupational therapy [44,45], while there is insufficient evidence for integrating them at the hospitals [46]. This could be another issue for future research.

We acknowledge some limitations of our scoping review. We followed most of the recommendations for scoping reviews of the PRISMA-ScR checklist [47]. However, we did not assess the quality of the studies, as is common in scoping reviews. We did not aim to research the efficacy of the interventions. In our research, we might have overlooked some studies that focused on diseases associated with delirium thus missing some relevant aspects in this field, e.g., in relation to stroke.

## 5. Conclusions

In conclusion, this review contributes to raising awareness of the medical and non-medical community of how to appropriately manage these conditions and could trigger research in this field. Both professions have emerging roles in the prevention and treatment of delirium. Already existing knowledge about specific occupational and physical therapy interventions should be clearly described and tested in future trials. Occupational therapy and physiotherapy interventions might have great potential in the rehabilitation and treatment of both people with dementia and of people with delirium superimposed on dementia. Yet, in this respect, additional research is needed. It appears to be promising to focus rehabilitation interventions through a person-centered approach (occupational history, personalized and meaningful activities), encourage environmental modifications and pursue active involvement of caregivers in rehabilitation sessions to reduce behavioral disorders or agitation. Thus, it would be important to investigate whether already established and effective occupational therapy programs known to reduce the behavioral and psychological symptoms in people with dementia [39,48] are also useful for the management and treatment of delirium and DSD. Regarding physiotherapy, on the other hand, it is crucial to learn about the amount and timing of intervention required to approach a person with dementia and/or delirium [49] and how to educate and train the caregiver. Finally, interdisciplinary collaboration among team members is strongly needed to improve the care of older people with delirium or delirium superimposed on dementia. In this new frontier for physiotherapy and occupational therapy, studies regarding the effectiveness of delirium prevention and treatment are necessary.

## Figures and Tables

**Figure 1 geriatrics-08-00028-f001:**
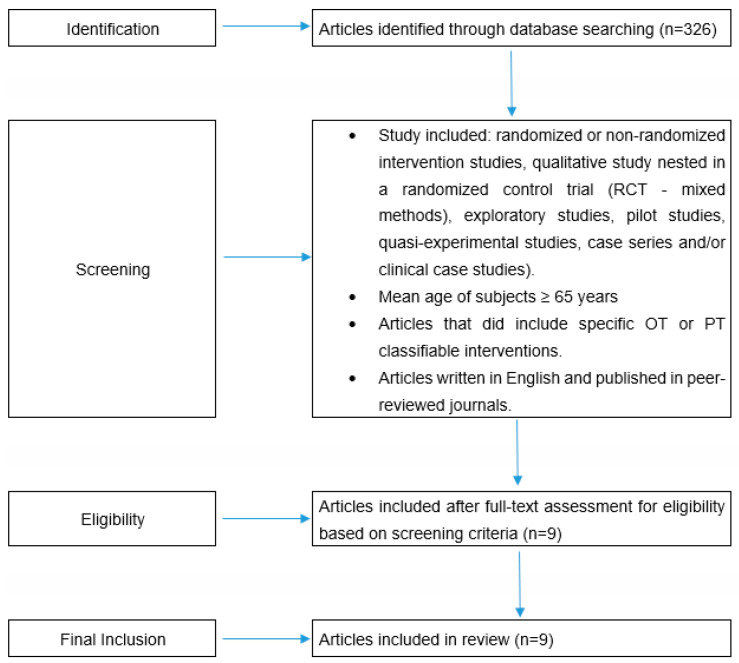
Study flow chart.

**Table 1 geriatrics-08-00028-t001:** Main characteristics and findings of the 9 articles selected for the review.

Author Year	Study Design	Setting	N	Mean Population Age	Diagnosis on Admission/Characteristics of the Cohort	Tools to Assess Delirium	Tools to Assess Dementia	Tools to Assess Functional Status	Tools to Assess Quality of Life for Patients and Caregiver	Rehabilitation Intervention	Results
Killington 2016 [9]	Qualitative study nested in a RCT	orthogeriatrics ward	28	87.5	Hip fractures	Not reported	MMSE	Not reported	Qualitative interview for caregivers and staff	- Mobilization - Activities of daily living	- Improvement of functional outcomes
Martinez-Velilla 2019 [10]	RCT	ACE Unit	370	87.1	Vulnerable population with a high level of functional reserve	CAM	MMSE	BI, SPPB handgrip strength	GDS EuroQoL 5D	- Balance and walking exercises at moderate intensity (twice daily) with PT - Resistance (2–3 sets of 8–10 exercises—30–60% load maximum equivalent) - Squats (standing up from a chair), leg press; bilateral knee extension - Active involvement of the caregiver - Patient education	- Improvement on SPPB (<0.001), Barthel Index (<0.001), MMSE (<0.001), GDS (<0.001), EuroQoL–5D (<0.001) and handgrip strength (<0.001) in experimental vs. usual-care group - Higher incidence of delirium in the experimental than usual care group (14.6% vs. 8.3%)
Said 2021 [11]	Single-blinded randomized controlled pilot trial	orthogeriatrics ward	51	83.5	Hip fractures	3D CAM SHORT CAM	MOCA IQCODE short form	Modified ILAS Gait Speed (m/s)	EuroQoL 5D EuroQoL-VAS	- Mobilization recumbent cycling in bed	- No functional improvement
Fraser 2015 [12]	Retrospective longitudinal study	Intensive Care Unit	132	64.7	Respiratory or gastrointestinal or cardiac or sepsis or neurologic problems	RASS CAM-ICU	Not reported	BI	Not reported	- Checking the correct positioning in bed every two hours - Generic exercises: sitting on the bed’s edge, standing, transferring to chair and walking	- Reduced rate of ICU readmission after 30 days (<0.001) - Reduced falls, sore pressures, adverse respiratory events (<0.001) - Barthel Index improvement (<0.001) - No reduction in days spent in ICU
Alvarez 2017 [13]	RCT	Intensive Care Unit	160	86.0	Sepsis; renal or hepatic failure; hemorraghe, acute respiratory syndrom, cardiac failure	CAM-ICU Delirium rating scale	MMSE	FIM Hand grip strength	Not reported	- Sensorial stimulation (i.e., lights, tactile body stimulation) - Mobilization - Cognitive stimulation - Passive and active exercises of the upper limbs - Family participation (twice a day)	- Decrease of delirium duration and functional improvement (FIM and handgrip strength)
Gonaan 2017 [14]	Prospective observational study	Dementia ward Hospital	30	80.3	Dementia or/and delirium	Not reported	MMSE	BI	Not reported	- Person-centered care (occupational story, tailored and meaningful activities) - Caregiver’s participation in the rehabilitation session - Staff education	- Decrease of behavioral disorders
Pozzi 2017 [15]	Case series	Rehabilitation Hospital	6	84.1	DSD	RASS	MMSE	BI Tinetti Scale	Not reported	- OT intervention (twice a day, 5 days, 40 min per day). - Sensorial and cognitive stimulation, meaningful occupations - Family education and involvement in the rehabilitation sessions - Environmental changes to promote rest, sleep-wake cycle improvement and reorientation	- Functional improvement - High rate (83%) of discharge to home
Pozzi 2020 [16]	Feasibility Study	Nursing home	22	86.4	DSD	m-RASS RADAR Delirium-O-Meter	MMSE CDR	BI Tinetti Scale	COPM (proxy)	- Caregiver education and involvement in the rehabilitation sessions - Multisensorial and cognitive stimulation with meaningful and tailored activities - Environmental changes to promote activities and rest	- Functional and occupational improvement from delirium diagnosis to resolution
Bolton & Loveard 2016 [17]	Case series	Hospice	88	Not reported	Dementia and/or delirium	Not reported	MOCA	Not reported	Qualitative interview for staff	- Meaningful and tailored activities - Environmental changes to improve orientation - Staff training to improve competence in the use of nonpharmacological strategies. - Caregiver education and involvement in the sessions	- OT was feasible in all patients - Patients’ agitation decrease

CAM (confusion assessment method); 3DCAM (3-Minute Diagnostic Interview for Confusion Assessment); CAM-ICU (Confusion Assessment Method Intensive Care Unit); RASS (Richmond Agitation and Sedation Scale); mRASS (modified Richmond Agitation and Sedation Scale); MMSE (Mini Mental State Examination); MOCA (Montreal Cognitive Assessment); ILAS (Iowa Level of Assistance Scale); IQCODE (Informant Questionnaire on Cognitive Decline in the Elderly); CDR (Clinical Dementia Rating); BI (Barthel Index); SPPB (Short Physical Performance Battery); FIM (Functional Independence Measure); GDS (Geriatric Depression Scale); EuroQoL VAS (EuroQoL Visual Analogue Scale); EuroQol 5D (EuroQoL questionnaire 5 dimensions); COPM (Canadian Occupational Performance Measure); RCT (randomized controlled trial); ACE (acute care for elders); ROM (range of motion).

## Data Availability

Not applicable.

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
