# Peer review of "Innovative Non-Pharmacological Management of Delirium in Persons with Dementia: New Frontiers for Physiotherapy and Occupational Therapy?"

_geriatrics, 2023, doi:10.3390/geriatrics8020028_

Round 1
Reviewer 1 Report
Introduction:
- Lines 53-56: please be more specific: do these lines apply to persons with delirium, dementia or both? In addition, in line 53 you describe that neuroleptics and benzodiazepines should be avoided in people at risk. At risk for what?
Methods:
- You describe that you focused on studies that examined the role of OT and PT in preventing and treating delirium, dementia and DSD in older adults. If I understand this sentence correctly, you were also interested in the role of OT and PT in persons with dementia without delirium. According to your search queries, you did not include a query with only dementia. You only included queries with delirium AND dementia, and with delirium only. Please explain.
Results:
- Table 1: please include reference numbers in your table, that makes it easier for readers to check your described findings.
- Line 97: total number is 887 and not 867.
- Line 98-101 and lines 105-108: you mention 8 of the 9 studies, please mention all studies.
- In lines 101-105 you mention that only four studies included persons with dementia, delirium or DSD. The other five studies included patients with a mix of medical conditions. Since the aim of your study was to explore the role of OT and PT in preventing and treating delirium, dementia and DSD in older adults, I suppose that these last five studies investigated the role of OT or PT on preventing delirium, or that these studies included a substantial amount of patients with dementia or delirium (to investigate the role of OT or PT on dementia or treating delirium). However, this does not become clear from the text or table. If my assumptions are wrong, then I do not understand why these studies were included. Suggestions:
o In your table, include which studies focused on preventing delirium (or DSD), treating delirium (or DSD), or on the role of OT or PT in dementia.
o In your table, include the number of persons with delirium, dementia or DSD.
- Lines 158-165: do these conclusions apply to persons with dementia, delirium, and/or DSD? Please be more specific.
Minor:
- Please use an English spelling checker. There are a few spelling mistakes, for example in your abstract (conclusion, line 33: effectivess) and discussion (line 175: physiotherapic).
Author Response
Thank you for the review.
We appreciated all your input. Attached is our letter with our responses.
Christian Pozzi

Reviewer 2 Report
Because of the growing percentage of elderly people in many societies a preventive therapy like occupational therapy and physiotherapy interventions of psychic geriatric symptoms is still only at a beginning stage. This is true especially for delirium and dementia. Regarding preventive therapy of these diseases the present literature review comes timely, is relevant and fills a gap in reviews about the outcome of these interventions. Such a comprehensive review is still lacking because the topics drug therapy or non-pharmacological therapies are only covered separately.
Because it is a scoping literature review a flow chart of the search depicted in a diagram would be helpful.
The conclusions are consistent, however, the authors should again mention some results of the studies screened, like “reduction of behavioral disorders or agitation by providing a person-centered approach (occupational story, tailored and meaningful activities), modifications of the environment and the active involvement of caregivers in rehabilitation sessions” as mentioned in results.
The references are appropriate.
Add flow chart of literature search - otherwise no figures are in the text.
Only some minor notes to improve the literature study:
please spell out ADL-activities (line 212);
please correct spacing error (line 195).
Author Response
Dear reviewer
Thank you for your review. We really appreciated your input. Attached are our responses to your suggestions.
Christian Pozzi
